# GG-Editor: Locally Editing 3D Avatars with Multimodal Large Language Model Guidance

Yunqiu Xu
ReLER Lab, CCAI
Zhejiang University
Hangzhou, China
imyunqiuxu@gamil.com

Linchao Zhu
ReLER Lab, CCAI
Zhejiang University
Hangzhou, China
zhulinchao@zju.edu.cn

Yi Yang*
ReLER Lab, CCAI
Zhejiang University
Hangzhou, China
yangyics@zju.edu.cn

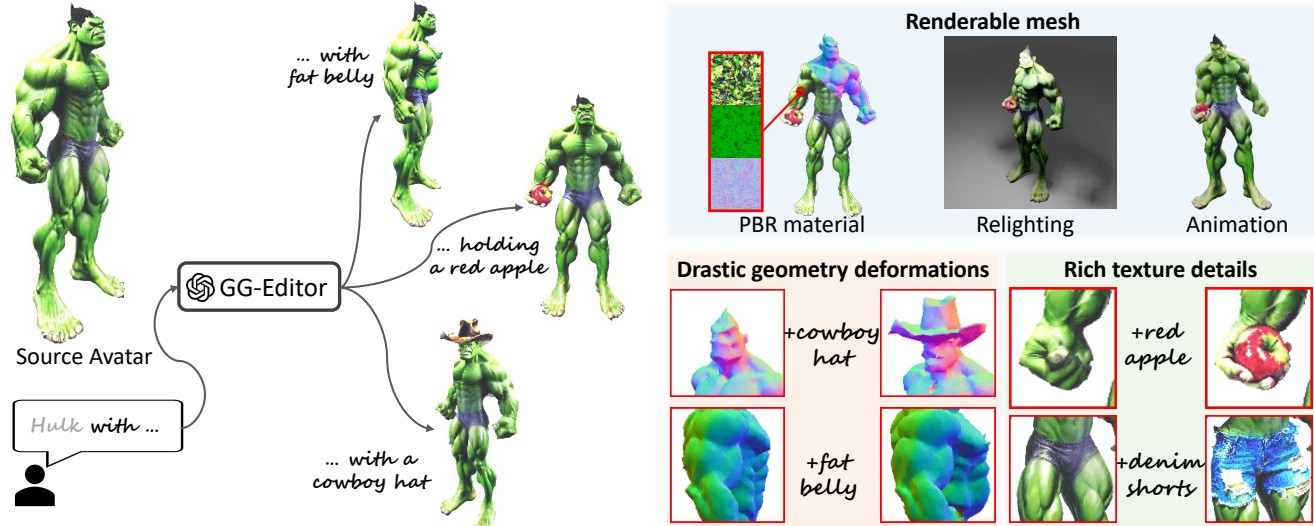

Figure 1: Given a source avatar and only text prompts, GG-Editor produces realistic local editing results drastic geometry deformations and rich texture details.

## Abstract

Text-driven 3D avatar customization has attracted increasing attention in recent years, where precisely editing specific local parts of avatars with only text prompts is particularly challenging. Previous editing methods usually use segmentation or cross-attention masks as constraints for local editing. Although these masks tightly cover existing objects/parts, they may limit editing methods to create drastic geometry deformations beyond the covered contents. From a different perspective, this paper presents a GPT-guided local avatar editing framework, namely GG-Editor. Specifically, GG-Editor progressively mines more reasonable candidate editing regions via harnessing multimodal large language models which already organically assimilate common-sense human knowledge. In order to improve the editing quality of the local areas, GG-Editor explicitly decouples the geometry/appearance optimization, and adopts a global-local synergy editing strategy with GPT-generated local prompts. Moreover, to preserve concepts residing in source avatars, GG-Editor proposes an orthogonal denoising score that orthogonally decomposes editing directions and introduce an explicit term for preservation. Comprehensive experiments demonstrate that GG-Editor with only textual prompts achieves realistic and high-fidelity local editing results, significantly surpassing prior works. Project page: https://xuyunqiu.github.io/GG-Editor/.

## CCS Concepts

• **Computing methodologies** → **Image manipulation**; **Computer vision**.

## Keywords

Text-Driven Editing; Diffusion Models; 3D Human Avatar Editing; Multimodal Large Language Models; Local Region Manipulation

**ACM Reference Format:**
Yunqiu Xu, Linchao Zhu, and Yi Yang. 2024. GG-Editor: Locally Editing 3D Avatars with Multimodal Large Language Model Guidance. In *Proceedings of the 32nd ACM International Conference on Multimedia (MM '24), October 28-November 1, 2024, Melbourne, VIC, Australia.* ACM, New York, NY, USA, 10 pages. https://doi.org/10.1145/3664647.3681039

*Corresponding author.

# 1 Introduction

Text-driven 3D avatar generation [26, 27, 41, 71, 74, 78] and editing [5, 20, 49, 60, 73] are crucial for various applications and industries, which enable the creation of customized digital humans using a few words. Despite notable advancements in 3D editing [8, 19, 21, 57, 79], locally editing some specific parts of 3D contents remains challenging, due to the ambiguity in language and the complexity of 3D space. In training, the editing-related regions should be specified and properly manipulated, while the irrelevant contents are maintained.

Prior local editing methods usually locate the candidate editing regions via manual designation (*e.g.*, sketch) [51], cross-attention attribution [59, 80] or open-vocabulary grounding/segmentation [33, 63] with manually selected text queries. Manually specifying an editing region or selecting a suitable text query for grounding it could be cumbersome and inflexible. Furthermore, it is difficult to ground some concepts that are not in current images with text queries, as shown in Fig. 3a. The editing regions indicated by attention/segmentation maps usually tightly cover existing objects within given images. These object/part masks may not be a good constraint for editing, since there could be a misalignment between the candidate editing regions and the object/part masks. For instance, the optimal region to add a hat is the area above the head, rather than the head region itself. Consequently, it is hard to introduce some non-rigid edits (*e.g.*, drastic geometry deformations), if the assigned editable region is the head region itself.

Motivated by these observations, we try to mine more reasonable editing regions using multimodal large language models (LLMs) [1, 3, 4, 7, 29, 42, 45, 47, 52, 66, 69] which already organically assimilate common-sense human knowledge. In this paper, we exploit the extraordinary text parsing and spatial reasoning capability of multimodal LLMs (*i.e.*, GPT-4V [1]), and present a GPT-guided 3D avatar editing framework, namely GG-Editor. In contrast to the existing LLM-guided generation methods [39, 40, 77], our editing method also requires a good understanding of the existing visual contents and reasoning in 3D space. To enable multimodal LLMs to handle 3D inputs and locate reasonable regions, we decouple the region seeking process into multiple steps: representative view selection, coarse grid region selection and iterative fine region mining. In addition, we inject some domain knowledge regarding avatars and devise various visual prompting strategies to enhance the grounding capability of GPT. As a result, GG-Editor progressively mines some local regions corresponding to the given editing prompts, and can use the mined regions for subsequent editing.

We incorporate the mined local editing regions into a geometry-appearance decoupled learning scheme [9, 53], where the geometry and appearance of the pre-trained source avatars are edited sequentially. We notice that using a standard human-centric camera pose sampling system may be less effective for local editing, as the local editing regions could be small or occluded in the rendered full-body images. To provide high-quality edits with more geometry and texture details, GG-Editor employs a global-local view synergy editing strategy that simultaneously renders images from global and focal views. Nevertheless, the semantics of the local-view images could be deviate from the given prompt describing the global avatar. To cope with such semantic misalignment issue, we leverage GPT to

analyze the source/target prompts, and then generate local prompts tailored for the local view.

Many prior 3D editing works [38, 80] optimize models using score distillation sampling (SDS) [55] with only target prompts. We believe that it is also important to exploit the information residing in source avatars and corresponding prompts. Drawing inspiration from an image editing approach [23], we treat the original contents as the reference and calculate the delta scores that steer the editing toward a less biased direction. We also observe that such delta score function may sometimes bring over-editing results that largely deviate from the source avatars. Thus, we orthogonally decompose the condition directions, and present a new orthogonal denoising score (ODS) loss that contains an explicit term to adjust the preservation of the original contents. In this way, GG-Editor brings well compositional and high-fidelity edits.

To the best of our knowledge, GG-Editor is the first multimodal LLM-guided framework for text-driven 3D avatar editing. We showcase it on multiple avatars with various editing prompts. Comprehensive experiments validate its superiority in locally editing avatars. The main contributions can be summarized as follows:

- We introduce a new GPT-guided framework for zero-shot text-driven 3D avatar editing, which first integrates common-sense human knowledge and progressively mines reasonable candidate regions for local editing.
- We devise a global-local view synergy editing strategy to improve the local editing results by training models with additional local renderings and GPT-generated local prompts.
- We present ODS loss that orthogonally decomposes the editing directions and introduces an explicit term to adjust the preservation of the source concept.

# 2 Related Works

***Controllable Text-Driven 3D Content Editing.*** Most current text-driven 3D editing methods globally manipulate (*e.g.*, style transfer) the whole scenes [21, 31, 34, 49, 65] or objects [10, 19, 20, 50, 57]. Implementing local editing and maintaining the unrelated areas is more challenging, which requires models to have a more fine-grained understanding of 3D contents and editing prompts. To improve the local editing controllability, some methods manually assign multi-view semantic sketches [51] or editable regions [6, 12, 36, 38] as auxiliary constraints. However, manually adding such constraints in a 3D space could be cumbersome and inflexible.

Numerous methods [11, 15, 22, 33, 63, 73, 76] try to generate masks as constraints for local editing, using off-the-shelf open-vocabulary segmentation or grounding methods [32, 43]. A few works [13, 14] attempt to locate the semantically related local regions on mesh surface using CLIP guidance [56]. Another line of work [28, 59, 79, 80] utilizes cross-attention mask-based techniques to obtain the editable regions. While these mask-based methods show promising results regarding rigid editing (*e.g.*, changing appearance), they usually struggle to bring drastic geometry deformations, as the editable regions have been aligned with existing objects/parts indicated by a manually selected text query. This paper, from a different perspective, attempts to mine reasonable editing regions with only text prompts and presents a local editing method effectively utilizing the obtained editable regions.

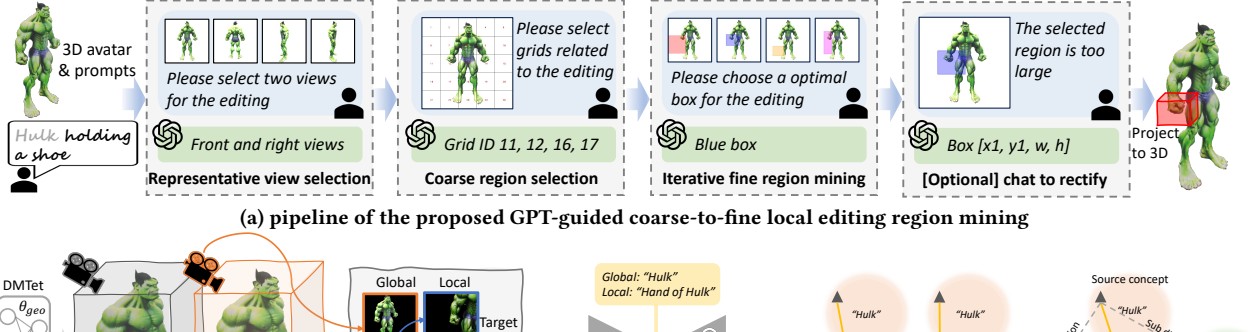

**(a) pipeline of the proposed GPT-guided coarse-to-fine local editing region mining**

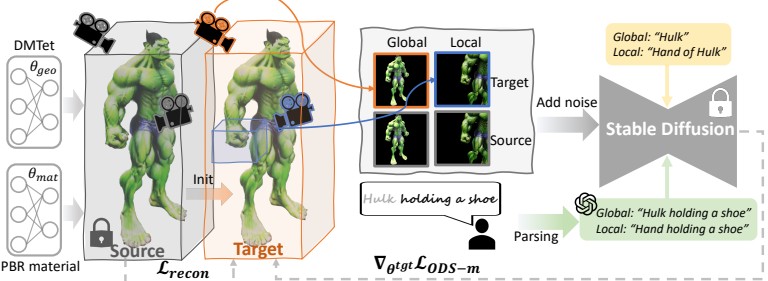

**(b) pipeline of the proposed geometry-appearance decoupled local editing with global-local view synergy**

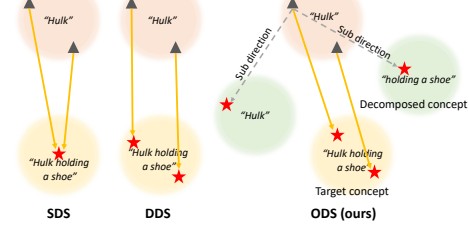

**(c) illustration of different editing processes using SDS, DDS and the proposed ODS**

**Figure 2: Overview of the proposed GG-Editor. With only textual prompts, GG-Editor first mines reasonable candidate regions for local editing. Using the candidate editing regions as constraints, GG-Editor then performs geometry-appearance decoupled local editing with global-local synergy and ODS loss.**

***LLM-Guided Visual Content Generation.*** LLMs, like GPT [1] and BLIP [37] series, have exhibited outstanding efficacy in many text-related tasks. Lv et al. [46] and Sun et al. [64] utilize LLMs to generate Python scripts commanding 3D software (*e.g.*, Blender) for text-to-video and 3D generation. Gao et al. [18] use GPT to generate scene graphs assisting compositional 3D scene generation. Another branch of research explores LLMs to generate various types of text-grounded layouts (*e.g.*, boxes and polygons) as constraints for text-to-image [16, 39, 70, 75], video [40, 44] and 3D generation [67, 77]. Unlike prior works that employ LLM to generate layouts that reflect the spatial relationships and motions described by given texts, we aim at exploiting multimodal LLMs to better understand both the textual and 3D visual inputs and then infer reasonable local regions for 3D editing.

## 3 Preliminary

***Geometry-Appearance-Decoupled 3D Representation.*** Learning 3D representations with explicit disentanglement of geometry and appearance has shown its effectiveness in 3D reconstruction [53] and text-to-3D generation [9]. In light of these findings, we adopt a similar two-stage scheme for editing.

In the first stage, DMTet [61] is utilized as the geometry representation $\theta_{geo}$, which can efficiently render high-resolution meshes with differentiable rasterization [35]. DMTet models 3D shapes using a deformable tetrahedral grid and an implicit SDF [54]. The SDF values and the position offsets of deformable tetrahedral vertices are learned using a MLP in DMTet. In training, the explicit mesh can be extracted through the differentiable marching tetrahedral layer.

Once the geometry model $\theta_{geo}$ is trained, an extra physically-based rendering (PBR) material model [48] is adopted to learn the

appearance representation. The material model $\theta_{mat}$ is parameterized using a MLP, which outputs diffuse value $k_d$, roughness and metallic value $k_{rm}$, and normal perturbation value $k_n$ for any point on the mesh surface extracted from $\theta_{geo}$. When both representations are optimized, we can produce textured meshes that are compatible with standard 3D tools and game engines.

***Score Distillation Sampling.*** SDS proposed by Poole et al. [55] has become a popular way to distill the diffusion priors for text-driven 3D generation and editing. Formally, given a diffusion model $\phi$ and images $\mathbf{x} = g(\theta, p)$ generated with a differentiable renderer $g(\cdot)$ and a camera pose $p$, SDS minimizes the difference between the added Gaussian noise $\epsilon$ and the predicted noise $\epsilon_\phi^s$:

$$\nabla_\theta \mathcal{L}_{SDS} = w(t) \left( \epsilon_\phi^s(\mathbf{z}_t; y, t) - \epsilon \right) \frac{\partial \mathbf{x}}{\partial \theta}, \tag{1}$$

where $y$ indicates the text condition and $\mathbf{z}_t$ is obtained by adding noise $\epsilon$ to $\mathbf{x}$ corresponding to the $t$-th timestep of the diffusion process. $w(t)$ denotes a weighting function determined by the time step $t$, and $\epsilon_\phi^s(\mathbf{z}_t; y, t)$ is the classifier-free guidance (CFG) [25]:

$$\epsilon_\phi^s(\mathbf{z}_t; y, t) = \epsilon_\phi(\mathbf{z}_t; \varnothing, t) + s \left( \epsilon_\phi(\mathbf{z}_t; y, t) - \epsilon_\phi(\mathbf{z}_t; \varnothing, t) \right), \tag{2}$$

where $\varnothing$ is a null condition. The conditioned prediction $\epsilon_\phi(\mathbf{z}_t; y, t)$ of the noise is extrapolated away from the unconditioned prediction $\epsilon_\phi(\mathbf{z}_t; \varnothing, t)$ by an amount controlled by a scalar $s$. During training, the diffusion model $\phi$ is frozen and the gradients are back-propagated to the parameterizable 3D representation $\theta$.

## 4 Methodology

Given a 3D avatar along with a text prompt describing the original visual content (*i.e.*, source prompt), our goal is to locally edit the avatar using another target prompt specifying the content after editing. It requires imposing proper manipulation on specific local

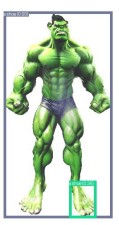 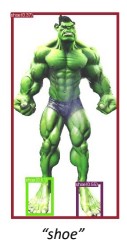

"holding"   "holding a shoe"   "shoe"

**(a) regions queried using Grounding DINO**

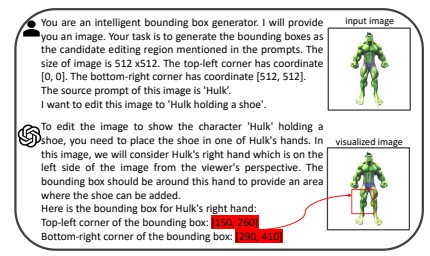

**(b) regions directly predicted by GPT**

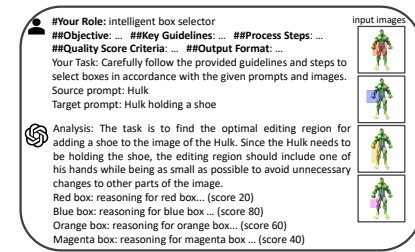

**(c) regions comparatively selected by GPT**

**Figure 3: Comparison of different approaches for mining the candidate local editing regions.**

regions corresponding to the editing prompts and retaining other editing-extraneous regions. Furthermore, the edited avatars should look realistic and keep cross-view consistency.

To achieve the above objectives, we propose GG-Editor a new GPT-guided avatar editing framework (see Fig. 2). Given an avatar with source and target prompts, we utilize multimodal LLMs (*i.e.*, GPT-4V) to analyze and gradually seek reasonable regions for editing. We devise a coarse-to-fine pipeline with various prompting strategies to alleviate the hallucination issue and obtain more accurate regions. The mined candidate editing regions are integrated into a geometry-appearance decoupled pipeline for local editing. For higher-quality local edits, a global-local synergy editing strategy is employed to optimize models with additional local views with GPT-generated local prompts. Moreover, we introduce an orthogonal denoising score, which performs 3D editing effectively and introduces an explicit term controlling the preservation of the source concept.

## 4.1 GPT-Guided Coarse-to-Fine Candidate Editing Region Mining

Unlike segmentation-based editing methods using accurate masks that cover existing contents, we aim to seek reasonable editing regions beyond given avatars with multimodal LLM guidance. Locating 3D local editing regions indicated by source/target prompts is quite difficult, though the powerful GPT is leveraged. We split the overall process into several steps and introduce a coarse-to-fine pipeline, gradually mining the candidate editing regions.

***Representative View Selection.*** To enable GPT to understand the 3D contents, we first project 3D avatars to 2D space. We empirically render four images from the orthogonal views (*i.e.*, front, back, left and right views) that are informative enough to represent the input meshes. Then, we request GPT to select a pair of images from the rendered images, where the selected image pair should be view-orthogonal and can better present the candidate edits.

Drawing inspiration from chain-of-thought [68], we also employ reasoning before answering prompting strategy to improve the robustness of multimodal LLMs. Specifically, we prompt GPT to first provide the descriptions of the given images and candidate edits, as well as the reasoning for the decision making before returning the selected views.

***Coarse Region Mining by Grid Selection.*** With the selected two renderings, we attempt to locate the regions relevant for candidate editing. However, we notice that GPT usually struggles with producing accurate coordinates, especially at fine granularity, as

shown in Fig. 3b. Thus, we devise several visual prompting [62, 72] strategies to unleash its locating capability.

We encode visual prompts to help GPT identify different regions within images. Concretely, we divide the images into different grids and assign different identifiers to each grid. We ask GPT first to analyze the required image modifications based on given source/-target prompts and then select several grids within the encoded images that are most relevant. The selected grids roughly indicate the regions for editing.

***Fine Region Mining via Iterative Verification.*** The fine region mining is formulated as an iterative proposal selection process. At each round of selection, one optimal box is selected as the reference, and then we jitter the coordinates of the reference box to generate multiple box proposals around the reference and avatar for the next round. At the beginning of selection, we initialize the reference box proposal based on the selected grids.

To enable GPT to be aware of different regions, we encode visual prompts in the rendered images by indicating these boxes with different colors, as shown in Fig. 3c. We also provide GPT with some key guidelines as well as the process steps for selecting a good proposal for editing. At each round of selection, GPT first interrupts the editing task and analyzes some key factors, based on the source/target prompts. GPT describes the regions covered by each box and scores each region with the corresponding reasoning before returning the final selection. Since the selected proposals could be noisy, we introduce a chain-of-verification strategy to alleviate the hallucinations of GPT. Concretely, the selected boxes in different rounds are stored, and we verify the previously selected boxes every $N_{ver}$ rounds. After multiple rounds of selection, we can obtain a reasonable candidate region for local editing.

Optionally, if the mined region does not meet the user's specific requirements, we can also interactively chat with GPT to further rectify the box to determine a better editing region. After obtaining the coordinates of two dimensions (*e.g.*, X-axis and Y-axis) from the first view, we perform a similar coarse-to-fine mining process on the other view using the fixed Y-axis coordinates to obtain the coordinates in another dimension (*e.g.*, Z-axis). We project all mined coordinates back to 3D space and generate a 3D bounding box indicating the editable regions.

## 4.2 Geometry-Appearance Decoupled Local Editing with Global-Local View Synergy

***Geometry and Appearance Editing with Local Constraints.*** As in the geometry-appearance decoupled framework discussed in

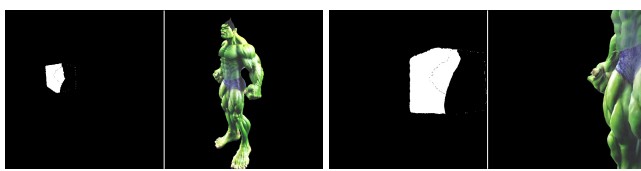

(a) renderings in global view $p^{glb}$ (b) renderings in local view $p^{loc}$

**Figure 4: Visualizations of the mined editable regions and images rendered from global and local views.**

Sec. 3, we sequentially manipulate the geometry and appearance of the given avatars. We first initialize the geometry and appearance models of the target avatar using the pre-trained source avatar: $\{\theta_{geo}^{tgt}, \theta_{mat}^{tgt}\} \leftarrow \{\theta_{geo}^{src}, \theta_{mat}^{src}\}$. Then, we successively optimize the target avatar's geometry $\theta_{geo}^{tgt}$ and appearance $\theta_{mat}^{tgt}$ using similar losses while keeping the source avatar fixed.

With a randomly sampled camera pose, we simultaneously render both the source and target avatars as well as the 3D bounding box indicating the candidate editing regions. Specifically, we calculate the editable map $\mathbf{m}$, based on the rendered depth map $\mathbf{d}^{box} \in [0, 1]$ and object mask $\mathbf{o}^{box} \in [0, 1]$ of the 3D box and the source avatar's depth map $\mathbf{d}^{src}$:

$$\mathbf{m} = \mathbb{1}_{\{\mathbf{d}^{box} \leqslant \mathbf{d}^{src}\}} \mathbf{o}^{box}, \qquad (3)$$

where $\mathbb{1}_{\{\mathbf{d}^{box} \leqslant \mathbf{d}^{src}\}}$ is the indicator function, being 1 if $\mathbf{d}^{box} \leqslant \mathbf{d}^{src}$ and 0 otherwise. To maintain the original contents, in both geometry and appearance editing stages, we impose reconstruction losses on the non-editable regions:

$$\mathcal{L}_{recon} = (1 - \mathbf{m})(\|\mathbf{o}^{tgt} - \mathbf{o}^{src}\|_2^2 + \|\mathbf{x}^{tgt} - \mathbf{x}^{src}\|_1), \qquad (4)$$

where $\mathbf{x} \in \{\mathbf{n}, \mathbf{c}\}$ denotes the normal/shading in geometry/appearance editing stages respectively.

Similar to SDS loss in Eqn. (1), we can distill the prior knowledge from a pre-trained text-to-image diffusion model [58] to the editable regions $\mathbf{m}$ for editing target avatar's geometry/appearance:

$$\nabla_{\theta^{tgt}} \mathcal{L}_{SDS-m} = \mathbf{m} w(t) \left( \epsilon_\phi^s(\mathbf{z}_t^{tgt}; y^{tgt}, t) - \epsilon \right) \frac{\partial \mathbf{x}^{tgt}}{\partial \theta^{tgt}}, \qquad (5)$$

where $y^{tgt}$ is the target prompt, and $\mathbf{z}_t^{tgt} \in \{\mathbf{z}_t(\mathbf{n}^{tgt}), \mathbf{z}_t(\mathbf{c}^{tgt})\}$ represents the noisy normals/shadings. $\theta^{tgt} \in \{\theta_{geo}^{tgt}, \theta_{mat}^{tgt}\}$ indicates the parameterizable 3D representation of avatar's geometry/texture.

***Global and Local Synergy with GPT-Generated Local Prompt.*** As the editable regions are determined, we present a global-local viewpoint synergy strategy to improve the editing quality of the local regions. Besides randomly sampling camera poses $p^{glb}$ around the whole body as in [9], we further set up another spherical coordinate system centered on the 3D box and sample some focal views $p^{loc}$ around the local editing regions. At each optimization step with Eqns. (4) and (5), we simultaneously render normals/shadings and editable masks from global and local views (see Fig. 4) to enrich geometry/texture details within local editing regions.

However, using shared prompts for both global and local views may not be optimal in training, because the semantics of local parts may differ from that of global views. As shown in Fig. 2b, using a prompt "Hulk holding a shoe" does not accurately describe the contents covered a local view that covers only the hand regions of

the Hulk. To cope with this issue, we harness the reasoning capability of GPT to parse global source and target prompts, and infer the local prompts for training. Specifically, given the source and target prompts, we ask multimodal LLMs to analyze the candidate editing areas as well as the interactions with the avatars, and then generate more appropriate prompts (*e.g.*, "hand holding a shoe") for local views.

## 4.3 Orthogonal Denoising Score

SDS loss is initially designed for text-to-3D generation and has become a common practice in many 3D editing works [38, 59, 80]. However, we believe it may not be optimal for editing tasks. As shown in Fig 2c, SDS loss leads samples from different views to one concept center defined by target prompts. We argue that it is also important to exploit the information residing in the original inputs (*i.e.*, source avatar and corresponding prompt) for 3D avatar editing.

Drawing inspiration from an image editing method [23], we use the images rendered from the source avatar as the reference to help the optimization. We add the identical noise $\epsilon$ to source and target inputs, and calculate the delta denoising score (DDS) loss:

$$\nabla_{\theta^{tgt}} \mathcal{L}_{DDS-m} = \mathbf{m} w(t) \left( \epsilon_\phi^s(\mathbf{z}_t^{tgt}; y^{tgt}, t) - \epsilon_\phi^s(\mathbf{z}_t^{src}; y^{src}, t) \right) \frac{\partial \mathbf{x}^{tgt}}{\partial \theta^{tgt}}. \quad (6)$$

Following the assumption introduced by Katzir et al. [30], we decouple the CFG score $\epsilon_\phi^s(\mathbf{z}_t; y, t)$ in SDS loss into three components:

$$\nabla_\theta \mathcal{L}_{SDS} = w(t) \left( \underbrace{\epsilon_\phi(\mathbf{z}_t; \varnothing, t)}_{\delta_N + \delta_D} + s \underbrace{(\epsilon_\phi(\mathbf{z}_t; y, t) - \epsilon_\phi(\mathbf{z}_t; \varnothing, t))}_{\delta_C} - \epsilon \right) \frac{\partial \mathbf{x}}{\partial \theta}, \qquad (7)$$

where $\delta_C$ is the condition direction leading the generated image towards the text condition $y$. $\delta_D$ and $\delta_N$ are domain correction and denoising direction respectively, which are not directly related to the editing prompts. Since the editing is from a pre-trained avatar that can render in-domain images, $\delta_D$ component is not effectively required and can be dropped. The noisy residual $\delta_N - \epsilon$ is relatively negligible and also can be dropped. Consequently, Eqn. (6) can be reformulated as:

$$\nabla_{\theta^{tgt}} \mathcal{L}_{DDS-m} = \mathbf{m} s w(t) \left( \delta_C^{tgt} - \delta_C^{src} \right) \frac{\partial \mathbf{x}^{tgt}}{\partial \theta^{tgt}}. \qquad (8)$$

Though optimizing models using the delta of source and target condition directions improves the editing effectiveness, it is still prone to bring edits that deviate significantly from the source avatars. To tackle this issue, we try to disentangle the editing direction, inspired by Perp-Neg [2]. Specifically, we orthogonally decompose the condition directions and introduce an explicit term to preserve source-concept contents, as shown in Fig. 2c. Suppose the projection and perpendicular of $\delta_C^{tgt}$ on $\delta_C^{src}$ are:

$$\Delta_{proj} = \frac{\langle \delta_C^{tgt}, \delta_C^{src} \rangle}{\|\delta_C^{src}\|^2} \delta_C^{src} \qquad \text{and} \qquad \Delta_{prep} = \delta_C^{tgt} - \Delta_{proj}, \qquad (9)$$

we present a new orthogonal denoising score (ODS) function and use it in both geometry and appearance editing stages:

$$\nabla_{\theta^{tgt}} \mathcal{L}_{ODS-m} = \mathbf{m} s w(t) \left( \lambda_{proj} \Delta_{proj} + \Delta_{prep} \right) \frac{\partial \mathbf{x}^{tgt}}{\partial \theta^{tgt}}, \qquad (10)$$

where $\lambda_{proj}$ is an adjustable term for source concept preservation.

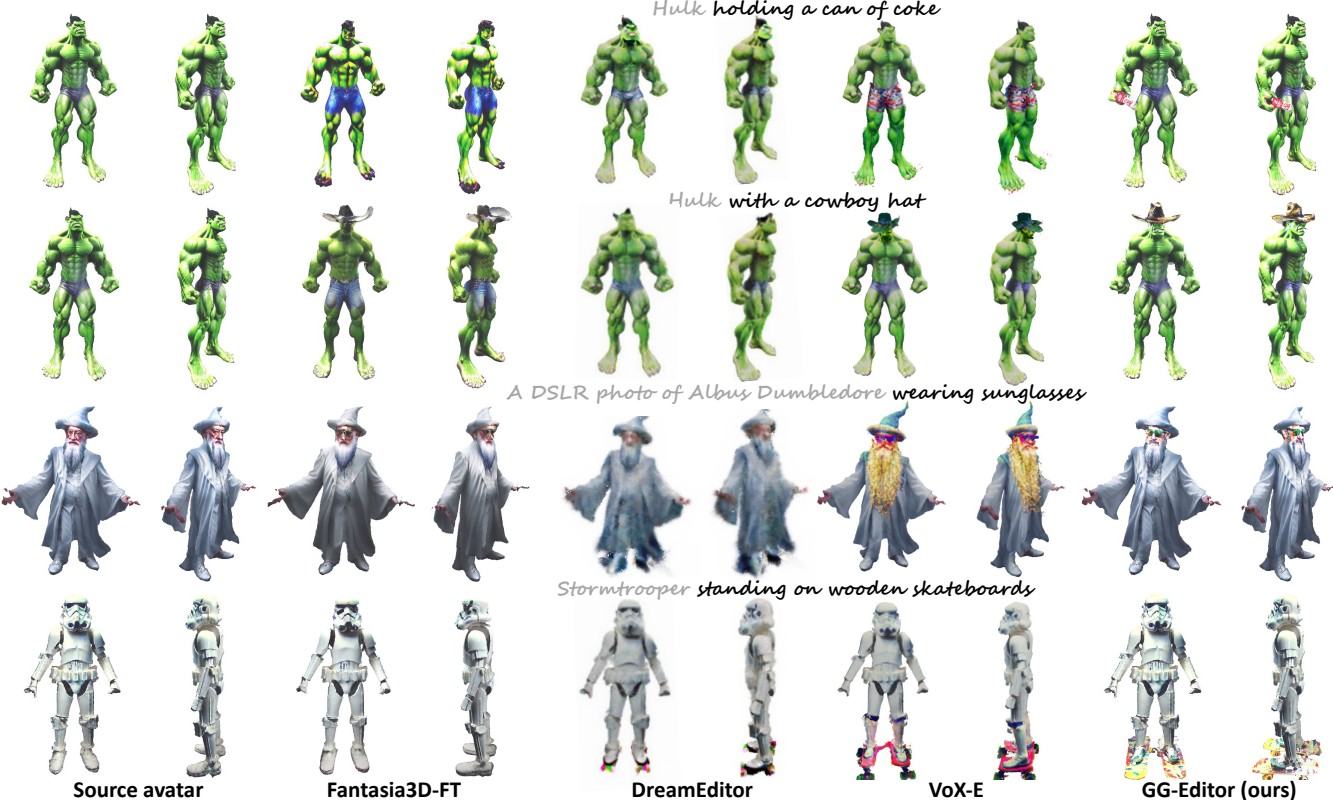

**Figure 5: Qualitative comparisons to three baselines on four different cases. The results show that our GG-Editor achieves realistic local editing results corresponding to the given prompts while better preserve the irrelevant regions. In addition, GG-Editor also shows high-quality results with richer geometry and texture details in the local editing regions. The source prompts are colored in gray.**

## 5 Experiments

***Implementation Details.*** We optimize our models with four NVIDIA RTX 3090 GPUs. For each avatar, we optimize the geometry and appearance models for 3K iterations (∼60 mins) and 2K iterations (∼40 mins) respectively. We adopt Stable Diffusion v1.5 as the diffusion prior for both geometry and appearance stages. AdamW optimizer is utilized with a learning rate of $10^{-3}$ and $10^{-2}$ for the two stages respectively. DMTet grid resolution is set to 128. The overall batch size is set to 4, where 2 for the local view. $N_{ver}$ and $\lambda_{proj}$ is empirically set to 5 and 0.2. The source avatars can be generated using Fantasia3D [9] or reconstructed using nvdiffrec [53].

***Evaluation Metrics.*** Following previous 3D editing works, we adopt CLIP similarity (CLIP$_{sim}$) [56], CLIP directional similarity (CLIP$_{dir}$) [17], Frechét Inception Distance (FID) [24] and peak signal-to-noise ratio (PSNR) for quantitative evaluation. CLIP$_{sim}$ measures the alignment between the target avatars and the target prompts. CLIP$_{dir}$ evaluates the alignment between the changes in both the avatars and text prompts. FID validates the edit magnitude, and PSNR quantifies the ability to preserve the source contents. Since the quality assessment of editing results could be subjective, we also conduct user studies for evaluation. Concretely, we provide 360° videos of source avatars and multiple target avatars edited by different methods, and ask users to select the best based on the local editing quality and the similarity to the source avatars.

**Table 1: Quantitative comparison with state-of-the-art methods on 16 cases.**

| | CLIP$_{sim}$↑ | CLIP$_{dir}$↑ | FID↓ | PSNR↑ | User↑ |
|---|---|---|---|---|---|
| Fantasia3D-FT [9] | 0.284 | 0.016 | 149.419 | 18.016 | 0.054 |
| DreamEditor [80] | 0.273 | 0.005 | 68.105 | **29.799** | 0.039 |
| Vox-E [59] | 0.289 | 0.021 | 102.845 | 26.178 | 0.093 |
| GG-Editor (ours) | **0.297** | **0.026** | **42.408** | 26.924 | **0.814** |

### 5.1 Main Results

We compare our GG-Editor to several recent advanced methods *i.e.*, Fantasia3D [9], DreamEditor [80] and Vox-E [59]. As Fantasia3D is a text-to-3D generation method that can be initialized with custom meshes, we adapt it to editing by fine-tuning pre-trained source avatars with target prompts and lower learning rate.

***Qualitative Comparisons.*** As shown in Fig. 5, without local region constraints, Fantasia3D-FT is prone to manipulate the entire avatars. DreamEditor and VoX-E can sometimes locate reasonable regions for local editing, but they usually show unrealistic results with limited geometry changes. Our GG-Editor successfully mines reasonable local editing regions and imposes faithful editing in relevant regions to the textual prompts, while the irrelevant regions are properly retained. We present more results of our GG-Editor in Fig. 6. Various high-quality normal and shaded RGB renderings validate the effectiveness and generalization of our local editing method. Please find more examples in the supplementary material.

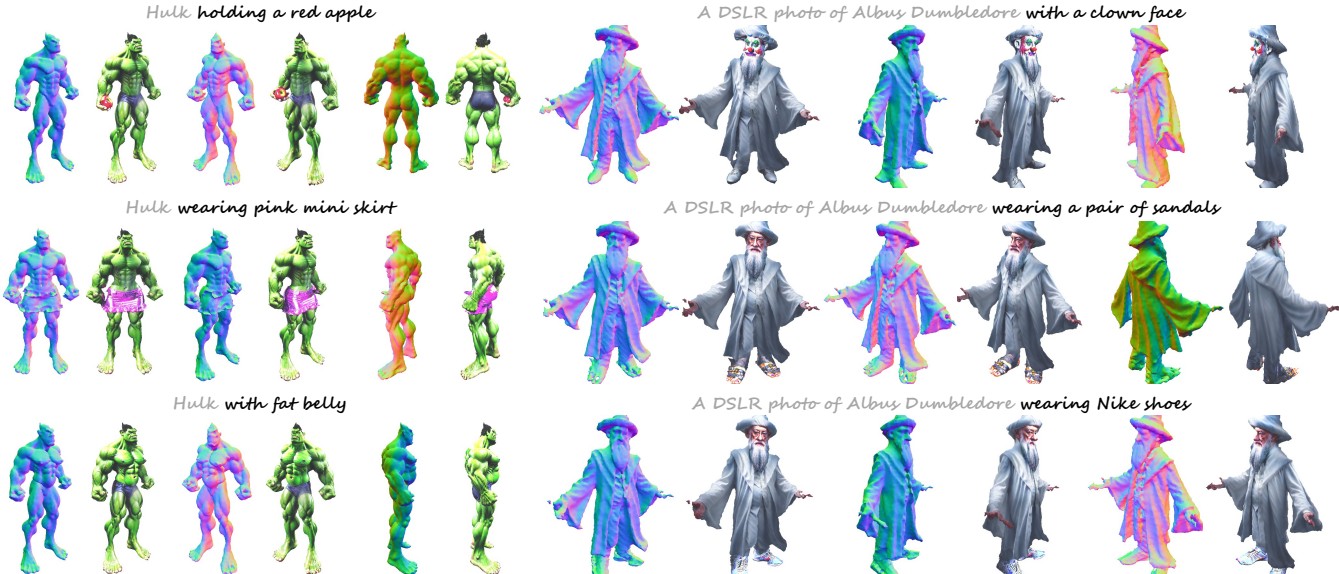

Hulk *holding a red apple*

A DSLR photo of Albus Dumbledore *with a clown face*

Hulk *wearing pink mini skirt*

A DSLR photo of Albus Dumbledore *wearing a pair of sandals*

Hulk *with fat belly*

A DSLR photo of Albus Dumbledore *wearing Nike shoes*

**Figure 6: More editing results of our GG-Editor. Both geometry and appearance results are visualized.**

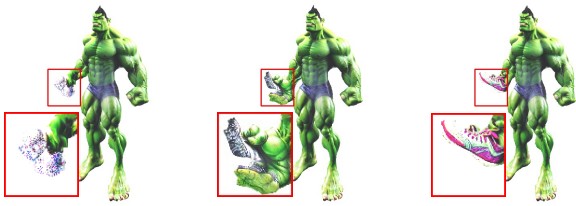

(a) w/o local view · (b) w/o local prompt · (c) w/ global-local syn.

**Figure 7: Ablation of global-local view synergy editing strategy. Our global-local view synergy editing strategy enhances local editing in terms of geometry and appearance.**

*Quantitative Comparisons.* We report the quantitative results of GG-Editor compared to several state-of-the-art methods in Table 1. Although GG-Editor manipulates a few local regions within source contents, it achieves the best results on $CLIP_{sim}$, $CLIP_{dir}$ and FID, demonstrating its excellent editing capability. Meanwhile, GG-Editor also obtains the second-best results on PSNR, indicating the strong ability to preserve the source contents. Moreover, GG-Editor receives 81.4% of the votes in user studies, which further validates the advantages of our local editing method from the perspective of human preferences.

## 5.2 Ablation Study

*Effectiveness of Global-Local Synergy.* To verify the necessity of the global-local view synergy strategy, we showcase an ablation on a challenging case (*i.e.*, *Hulk holding a shoe*) in Fig. 7. We compare it to the model trained using only global views and the model trained with global and local views but without the local prompts. In Fig. 7a, we can find blurry results are achieved when training without the local view. After adding the local views, edited results have more geometry and texture details. However, using a global prompt "*Hulk holding a shoe*" is not optimal to describe the local region around the hand, which could bring noise in the optimization process. In our global-local synergy, the local region can be specified by the GPT-generated local prompt, which facilitates the local editing.

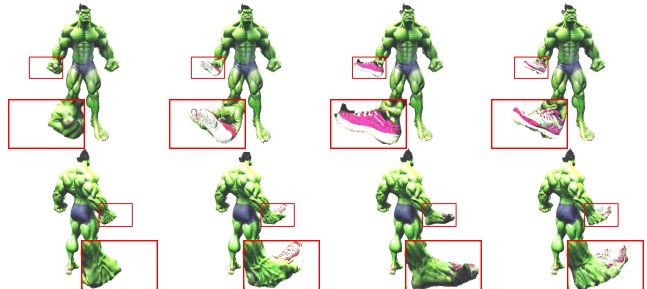

(a) source avatar · (b) w/ SDS loss · (c) w/ DDS loss · (d) w/ ODS loss

**Figure 8: Comparison of editing losses. Our ODS loss brings high-quality editing results (*i.e.*, shoe), while preserves the source concept (*i.e.*, hand).**

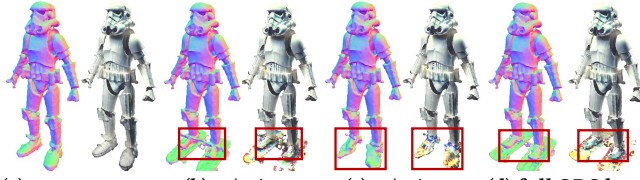

(a) source avatar · (b) w/o $\Delta_{proj}$ · (c) w/o $\Delta_{prep}$ · (d) full ODS loss

**Figure 9: Analysis of the projection and perpendicular terms in ODS loss. When the both terms are utilized, high-fidelity edits are achieved without artifacts.**

*Effectiveness of Orthogonal Denoising Score* As shown in Fig. 8, the shoe optimized using SDS is still unclear and unrealistic, while DDS and ODS effectively add a realistic shoe to the hand. Since the concept of the shoe is quite close to that of the foot, the Hulk's hand becomes a foot through optimization with DDS (see hand regions in the bottom images). In contrast, ODS can better preserve the source concept and retain the hand in the back view. In Fig. 9, We also analyze the effectiveness of orthogonally terms in ODS loss. Without projection and perpendicular terms, the edited avatars show obvious artifacts and ineffective editing respectively.

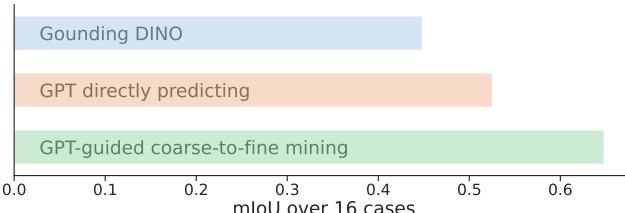

**Figure 10: Quantitative analysis of our proposed GPT-guided local region mining.**

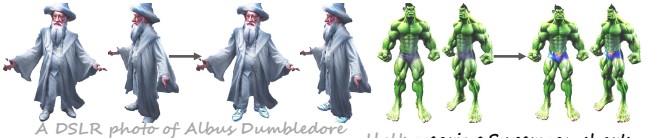

**Figure 11: Examples of appearance editing with fixed shapes.**

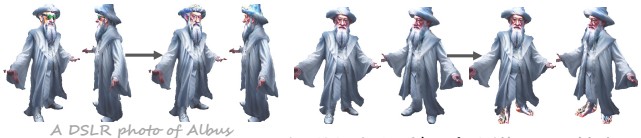

**Figure 12: Examples of removal editing using GG-Editor.**

***Effectiveness of GPT-Guided Editing Region Mining.*** In order to show the superiority of the proposed GPT-guided coarse-to-fine editing region mining approach, we measure the overlap of the mined local editing regions with respect to the human-preferred areas using mIoU. As shown in Fig. 10, compared to grounding DINO [43] and directly predicting coordinates via GPT, our proposed method can select candidate editing regions that are more consistent with human preferences.

## 5.3 More Applications

As the optimization process of geometry and appearance is explicitly decoupled, it is easy to retexture the avatars with fixed shapes, as shown in Fig. 11. Besides addition and modification editing, GG-Editor can also performs removal editing as illustrated in Fig. 12 Another characteristic of our proposed method is that GG-Editor can make local edits incrementally. Fig. 13 showcases an example of incremental editing. In each editing step, only the contents within the local editing regions are manipulated, while contents irrelevant to the editing are retained. In addition, GG-Editor directly exports textured avatar meshes that can be used for various downstream applications like relighting and animation in the classic graphics pipeline, as visualized in Fig. 14.

## 6 Limitations and Future Works

As a pioneer in taming multimodal LLMs for text-driven 3D local editing, GG-Editor presents realistic editing results. However, due to the limited performance of existing text-to-image diffusion models [58] for hand and human-object interaction generation, we find it sometimes fails to edit challenging cases faithfully (see left part of Fig. 15). As our method maintains the editing irrelevant contents by directly regressing the normals and colors of source avatars, our method could bring some artifacts around the boundaries of the local editing regions (see middle part of Fig. 15). Since GPT-4V lacks

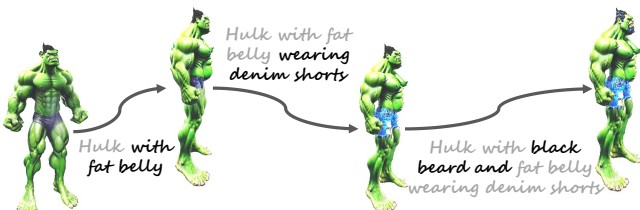

**Figure 13: Examples of incremental editing using GG-Editor.**

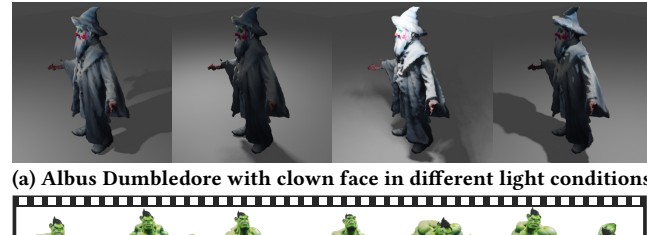

**(a) Albus Dumbledore with clown face in different light conditions**

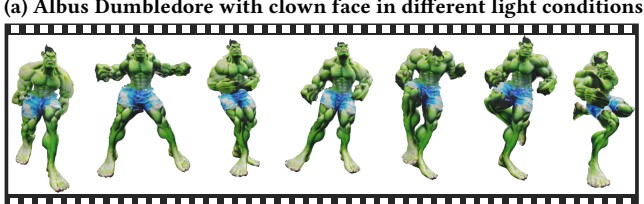

**(b) Hulk wearing denim shorts is doing swing dancing**

**Figure 14: Examples of relighting and animation of avatars edited using GG-Editor.**

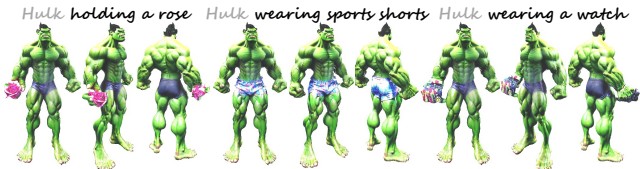

**Figure 15: Failure cases of our proposed method.**

accurate localization capacity, it is hard for GG-Editor to localize some tiny regions from avatars precisely, and such mislocalization may result in some unsatisfactory results (see right part of Fig. 15).

GG-Editor is an early attempt at 3D local editing with multimodal LLM guidance, focusing only on 3D human avatars. In the future, we would like to explore mining local editing regions from more challenging and general scenes. In addition, we will investigate enhancing the locating capability of multimodal LLMs, thereby improving the controllability of 3D local editing.

## 7 Conclusion

This paper proposes a new multimodal LLM-guided framework for locally editing 3D avatars, namely GG-Editor. GG-Editor harnesses GPT-4V combined with human common sense knowledge to infer some reasonable local editing regions beyond existing avatars. To enrich the geometry and texture details within local editable regions, we devise a global-local view synergy editing strategy. Integrating it into a geometry-appearance decoupled learning pipeline, GG-Editor achieves high-fidelity local editing results with cross-view consistency. Besides, we present ODS loss that orthogonally decomposes editing directions and introduces an explicit term for adjusting source concept preservation. Experiments with multiple avatars and various editing prompts showcase the effectiveness and superiority of our GG-Editor for local avatar editing.

# Acknowledgments

This work was supported in part by the Nature Science Foundation of China under Grant U2336212. This work was also partially supported by the Fundamental Research Funds for the Central Universities under Grants 226-2022-00051 and 226-2024-00058.

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
