# OpenReview forum: "GG-Editor: Locally Editing 3D Avatars with Multimodal Large Language Model Guidance"
_acmmm.org/ACMMM/2024/Conference — MM2024 Poster_

### Official Review · Reviewer_Cf1L · 2024-05-23

**Rating:** 4
**Confidence:** 3

**Summary:**

The paper introduces GG-Editor, a framework designed for locally editing 3D avatars, guided by multimodal large language models like GPT-4V. It allows precise edits to specific parts of the avatar using text prompts while leaving other parts unchanged. This framework can manage geometry deformations and maintain texture details. It identifies appropriate editing regions through a coarse-to-fine process and utilizes a global-local synergy editing strategy combined with an orthogonal denoising score  loss to improve editing quality. Experiments demonstrate that GG-Editor achieves realistic and high-fidelity local edits with only text prompts.

**Strengths:**

1. A framework called GG-Editor is proposed, which utilizes multimodal large language models to guide the local editing of 3D avatars.
2. GG-Editor can edit specific local areas of a 3D avatar based on text prompts while keeping the rest of the areas unchanged.
3. By employing a global-local synergy editing strategy, GG-Editor can enhance the geometric and textural details of the local editing areas while maintaining overall consistency.
4. An orthogonal denoising score loss is introduced, which aids in decomposing editing directions during the editing process and explicitly controlling the preservation of the original content.

**Limitations:**

1.	On line 440 of the paper, it states, "if the mined region does not meet the user's specific requirements, we can also interactively chat with GPT to further rectify the box to determine a better editing region." It seems that the process of GPT-Guided Candidate Editing Region Mining still requires human intervention. I feel that this method is somewhat over-engineered, and it might be simpler to manually specify the position of the 3D bounding box.
2.	The Orthogonal Denoising Score is an important innovation of the paper, but there are too few experiments regarding ODS in the paper. More detailed experiments are needed to illustrate the properties of ODS.
3.	The experiments are not comprehensive, and the variety of editing examples is limited.
4.	The paper extensively describes the use of prompt engineering to explore the potential of GPT. However, this approach may make it difficult to apply the method directly to a wide variety of different 3D avatars without human intervention.

**Suitability:**

3

---

### Official Review · Reviewer_P4mv · 2024-05-24

**Rating:** 2
**Confidence:** 4

**Summary:**

The paper introduces a MLLM-guided framework for text-driven 3D avatar editing. The framework consists of:
1. A novel MLLM-guided pipeline for deciding the reasonable region to edit.
2. A geometry-appearance decoupled editing strategy with global-local view synergy, which improves the local results with additional local renderings and generated prompts.
3. An orthogonal denoising score that decomposes the editing directions to preserve the information in the original source inputs.
Comprehensive experiments showcase the effectiveness of the framework for local avatar editing, achieving high-quality local editing results in either geometry or texture details with cross-view consistency.

**Strengths:**

1. The research problem of text-to-3D editing is challenging due to the ambiguity in language and the complexity of 3D space. This work would provide some insights into bridging the gap between these two modalities.
2. The idea of leveraging MLLM to decide the optimal editing regions and guide the content generation is novel and the corresponding method is rather intuitive and effective.
3. The experiments and demos are comprehensive and reader-friendly, displaying the effectiveness of your framework in different situations and the failure cases point out the limitations clearly.

**Limitations:**

1. About more demos: your work focuses on the "editing" of 3D avatars. The term "editing" generally involves three types of modifications: addition, replacement, and removal. In your demos, you have displayed your framework's capability in adding (e.g., Hulk with a cowboy hat / holding a can of coke) and replacing (e.g., Hulk in denim shorts) the semantics in both geometry and texture details, but not result regarding removing is displayed. I would be more convinced if more related demos (e.g., A DSLR photo of Albus Dumbledore with no hat) were provided.
2. About artifacts and failure cases: you have demonstrated some cases of your frameworks where the failures are rather obvious and ridiculous. Since the focus of your work is trying to leverage the capability of MLLM to the extent feasible, I believe this type of failure could be somehow mitigated by the powerful visual reasoning ability of MLLM. Yet you have not explored this field in your work.
3. About the editable regions: the editable regions generated in your framework is a 3D bounding box, which still seems not fine-grained enough in certain situations, for example, "A DSLR photo of Albus Dumbledore with a long magic wand (or walking stick)". A bounding box covering a long stick might take up a very large volume (even though the stick only takes up a small portion of room), resulting in the malfunction of the editing region mining pipeline. More inspection into this problem would make your work more comprehensive.
4. Typo and grammar check: in lines 158-159, "To coping with..." is more appropriate with "In coping with..." or "To cope with...".

**Suitability:**

3

---

### Official Review · Reviewer_Z7UG · 2024-05-27

**Rating:** 4
**Confidence:** 3

**Summary:**

This work uses GPT-4V to guide 3D avatar editing tasks. The geometry and texture of 3D avatar are represented by a DMTet ($\theta_{geo}$) and an MLP, respectively ($\theta_{mat}$). The editing process is divided into two steps:

1. Identifying the modification area: GPT-4V selects the appropriate editing area based on the prompt and maps the 2D editing area back to 3D.
2. Fine-tuning: Ensures consistency in unedited areas and introduces an Orthogonal Denoising Score to guide the fine-tuning process.

This work integrates several proposed methods, including Visual ChatGPT [1], FocalDreamer [2], and Delta Denoising Score [3]. Hence, the core innovation of thie work is not clear. Additionally, there are some weaknesses in Evaluation (see limitations). Nevertheless, the overall clarity and writing quality of the paper are high, leading to a borderline acceptance rating.

**Strengths:**

1. The article has a clear logic and high writing quality, making the overall approach easy to understand. It also includes a video demo.
2. GPT-4V is used to determine the 3D editing area based on 2D rendering results and well-designed prompts.
3. Compared to previous work, a source prompt and the the delta denoising score are introduced, with modifications tailored to the editing task.

**Limitations:**

1. Using multimodal LLMs and prompt engineering to analyze images is not new. Visual ChatGPT, EmbodiedGPT [4], and VisionLLM [5] have already excelled in this task. What is your innovation?
2. There is a lack of detailed explanation and experiment regarding the source prompt. How is the source prompt generated? How do different forms of source prompts affect editing results? It seems the purpose of introducing the source prompt is to justify the Orthogonal Denoising Score (ODS) in Section 4.3. What is the performance of your method if removing the the source prompt and ODS ?
3. Line 504 lacks a detailed explanation of the spherical coordinate system.
4. The comparative experiments are incomplete. I noticed that this paper cited FocalDreamer. Adding a comparison with FocalDreamer would enhance the experiment's credibility. Since FocalDreamer is not yet open-source, you could select a common 3D avatar (e.g., Flash Gordon or Turtle) from its project page, apply the same modifications, and present the results.
5. The user experiment lacks specific settings, such as the number of participants and gender ratio. Additionally, user experiments should allow participants to change viewpoints randomly to experience the 3D model, rather than just providing a video.


### References
[1] Wu, Chenfei, et al. "Visual chatgpt: Talking, drawing and editing with visual foundation models." *arXiv preprint arXiv:2303.04671* (2023).

[2] FocalDreamer: Text-Driven 3D Editing via Focal-Fusion Assembly, AAAI 2024

[3] Delta Denoising Score, ICCV 2023

[4] EmbodiedGPT: Vision-Language Pre-Training via Embodied Chain of Thought, NIPS 2023

[5] VisionLLM: Large Language Model is also an Open-Ended Decoder for Vision-Centric Tasks, NIPS 2023

**Suitability:**

2

---

### Meta-Review · Area_Chair_8Vxc · 2024-06-30

**Recommendation:** Accept (Poster)
**Confidence:** 4

**Metareview:**

All reviewers give positive ratings after the rebuttal. The AC agrees with the reviewers and recommend acceptance.